# The Role of Dendritic Cells in Immune Control and Vaccination against γ-Herpesviruses

**DOI:** 10.3390/v11121125

**Published:** 2019-12-05

**Authors:** Christian Münz

**Affiliations:** Viral Immunobiology, Institute of Experimental Immunology, University of Zürich, Winterthurerstrasse 190, CH-8057 Zürich, Switzerland; christian.muenz@uzh.ch

**Keywords:** Epstein Barr virus, Kaposi sarcoma-associated herpesvirus, plasmacytoid dendritic cells (pDCs), DEC-205, IL-12, type I IFN, adoptive T cell transfer

## Abstract

The two human oncogenic γ-herpesviruses, Epstein Barr virus (EBV) and Kaposi sarcoma-associated herpesvirus (KSHV), are prototypic pathogens that are controlled by T cell responses. Despite their ubiquitous distribution, persistent infections and transforming potential, most carriers’ immune systems control them for life. Therefore, they serve as paradigms of how near-perfect cell-mediated immune control can be initiated and maintained for decades. Interestingly, EBV especially quite efficiently avoids dendritic cell (DC) activation, and little evidence exists that these most potent antigen-presenting cells of the human body are involved in the priming of immune control against this tumor virus. However, DCs can be harnessed therapeutically to expand virus-specific T cells for adoptive transfer therapy of patients with virus-associated malignancies and are also currently explored for vaccinations. Unfortunately, despite 55 and 25 years of research on EBV and KSHV, respectively, the priming of their immune control that belongs to the most robust and durable immune responses in humans still remains unclear.

## 1. Epstein Barr virus and Kaposi Sarcoma-Associated Herpesvirus

The immune system is the organ of a multicellular organism that ensures a homeostatic balance between self (healthy cells) and non-self (beneficial commensal microbiota versus pathogens and sick cells) [1]. It has adapted during evolution in a species-specific manner by gene family expansions and contractions to environmental factors that challenge the majority of a species with grave morbidity and mortality prior to reproduction [2,3]. Among these are the human γ-herpesviruses Epstein Barr virus (EBV) or human herpesvirus 4 and Kaposi sarcoma-associated herpesvirus (KSHV) or human herpesvirus 8 which are widely distributed in the human population and threaten their host with tumor induction [4,5]. Indeed, EBV persistently infects more than 95% of human adults, and more than 70% are seropositive for KSHV in some areas of Sub-Saharan Africa, the cradle of humankind. Of the two viruses, EBV has the by far stronger growth transforming ability and can immortalize B cells in vitro [6]. In addition, it is associated with B cell lymphomas, such as Burkitt and Hodgkin lymphoma, natural killer (NK)/T cell lymphomas, epithelial cell-derived carcinomas, like nasopharyngeal carcinoma and gastric carcinoma, and smooth muscle tumors in patients [7]. Even so, KSHV cannot transform any cells in culture and does not sustain its own persistence without EBV co-infection [8,9,10]; it is associated with B cell lymphomas and the endothelial cell-derived Kaposi sarcoma in patients [11]. In one B cell lymphoma, namely primary effusion lymphoma (PEL), both EBV and KSHV co-infect the tumor cells in 90% of cases [12]. This is also the tumor entity that sustains KSHV after outgrowth in vitro. Thus, KSHV and EBV account for around 1–2% of all tumors in humans each [12], but their associated tumorigenesis is still incredibly rare in comparison to their distribution in the human population. 

In most carriers of these two oncogenic γ-herpesviruses, their premalignant infectious states are kept in check by the immune system [13]. This becomes apparent under conditions of immune suppression by human immunodeficiency virus (HIV) co-infection, iatrogenic immune suppression after organ transplantation, or primary immunodeficiencies, which lead primarily to EBV associated lymphomas and KSHV associated lymphomas or Kaposi sarcoma. The lesions in the human immune system of patients with primary immunodeficiencies identify T lymphocytes as the main components of the immune control of the two γ-herpesviruses, cytotoxic CD8^+^ T and NK cells for EBV and IFN-γ producing T cells for KSHV [13,14,15]. In contrast to the protective function of cytokines against KSHV, cytokine production due to T cell hyperactivation during ill-controlled EBV infection can lead to immune pathologies, including the symptomatic primary EBV infection, called infectious mononucleosis (IM) and hemophagocytic lymphohistiocytosis (HLH) [4,16,17]. In rare cases, these immune pathologies might also develop into autoimmune diseases, with the central nervous system (CNS)-affecting autoimmune disease multiple sclerosis (MS) being a likely candidate [18]. Thus, the human immune system has learned during its co-evolution with these two oncogenic γ-herpesviruses to efficiently control these abundant pathogens, and uses primarily T cell-mediated immune control without much contribution by antibodies for these tasks.

## 2. Priming of γ-Herpesvirus Specific T Cell Responses by Dendritic Cells in Preclinical Models 

This obviously begs the question how this near-perfect T cell-mediated immune control of EBV and KSHV is primed and if similar pathways could be used to initiate anti-tumor immune responses by vaccination. The main contestants for the priming of protective T cell immunity against EBV and KSHV are the infected B cells themselves and dendritic cells (DCs) that process fragments of dying, lysed, and lytically virus replicating cells for T cell priming. Along these lines, it has been shown in several studies over the years that EBV-transformed B cells (lymphoblastoid cell lines or LCLs) are 100-fold less efficient than DCs in stimulating T cells unless they use their receptor-mediated antigen uptake, including the B cell receptor and the decalectin DEC-205. Upon receptor-mediated antigen uptake by B cells, the gap in T cell priming efficacy narrows [19,20]. Furthermore, B cells primarily process and present antigen that is present in lymphoid tissues, while DCs can transport antigens from peripheral tissues for T cell priming in secondary and tertiary lymphoid structures [21]. Indeed the cross-presentation of EBV antigens by DCs, specifically MHC class I presentation of endocytosed fragments of EBV infected B cells, can be demonstrated in vitro both for memory T cell expansion and T cell priming [22,23]. However, in these studies, antigen cross-presentation was facilitated either by cell death induction in LCLs or by addition of a DC to an EBV-infected B cell ratio of 1:1. It is, however, more likely that DCs are not as abundant during the priming of EBV-specific T cell responses and also at sites to which EBV-infected B cells home to, namely germinal centers in lymphoid tissues. Indeed, very little conventional DC (cell-mediated) activation as measured by IL-12 production can be observed upon EBV infection of human peripheral blood mononuclear cells (PBMCs) or after even high dose EBV infection of mice with reconstituted human immune system components (HIS mice) [24]. In good agreement with very limited cDC involvement in CD8^+^ T cell priming during EBV infection, mutant EBV viruses that lack EBV-encoded small non-coding RNAs (EBERs) as the main pathogen-associated molecular pattern (PAMP) that was described to activate cDCs [25], replicate to the same viral loads in HIS mice and expand CD8^+^ T cells to the same extent as wild-type EBV. In contrast to cDCs, pDCs react to EBV infection of PBMCs [24,26,27]. They produce the type I interferon IFN-α17 in response to EBV exposure and are depleted during primary infection from peripheral blood of HIS mice and patients with IM [24,28,29]. However, Flt3-L induced pDC expansion does not increase CD8^+^ T cell responses to EBV infection in HIS mice, and, vice versa, pDC depletion does not compromise CD8^+^ T cell expansion [24]. Even daily injections of large amounts of IFN-α14 or IFN-α17 for the two first weeks of primary EBV infection in HIS mice only transiently suppress EBV viral loads and CD8^+^ T cell expansion [24]. Thus, EBV infection does not elicit significant IL-12 production by cDCs, and the readily detectable pDC activation does not alter viral loads nor T cell activation. Thus, DCs might be redundant for the priming of protective T cell-mediated immune control of EBV.

## 3. Evidence for the Involvement of Dendritic Cells in the Initiation of EBV and KSHV Specific Immune Control in Natura

These limited effects of DCs during EBV infection are also reflected in primary immunodeficiencies that indicate a cDC or pDC involvement. Interleukin 12 (IL-12) is the prototypic cytokine for Th1 polarized cell-mediated immune control induction by cDCs against intracellular pathogens [30]. It is particularly connected to type II interferon (IFN-γ) induction in T and NK cells. Deficiencies in IFN-γ production in response to IL-12 production (mutations in IL-12p40, IL-12 receptor β1, IRF8, ISG15, and NEMO) and in IFN-γ detection (mutations in IFN-γ receptor 1 and 2, STAT1, IRF8, and CYBB) do not predispose for uncontrolled EBV pathology [14], but instead sensitize for mycobacterium associated pathologies [31]. In contrast, IFN-γ receptor 1 and STAT4 deficiencies lead to KSHV pathology [32,33]. However, in particular, endothelial cell infection by KSHV leading to Kaposi sarcoma seems to be impaired. Thus, cDC mediated IL-12 production is not required for the induction of EBV specific immune control, and this virus does not seem to significantly trigger cDC maturation in the first place, while during KSHV infection, IL-12 mediated IFN-γ production seems to be mainly required to control this virus in endothelial cells.

In contrast to cDCs, pDCs are stimulated by EBV, but also primary immunodeficiencies that compromise their type I IFN production or sensing of this group of cytokines are not associated with EBV pathology [14]. These include mutations in IFN-α/β receptor, STAT1, IRF7, and TYK2 and predispose for α-herpesvirus, e.g., herpes simplex virus (HSV) and associated encephalitis [34]. This suggests that similar to our in vivo EBV infection studies in HIS mice, type I IFN and its main source pDCs make only a minor contribution to EBV specific immune control.

Instead, cytotoxic lymphocyte development, stimulation, expansion, and effector functions are required to control EBV and possibly also KSHV [13,14]. Cytokine production, e.g., IFN-γ, in the absence of cytotoxic lymphocyte function and then often even exacerbated by uncontrolled viral replication rather leads to immunopathologies like HLH [17]. Therefore, primary immunodeficiencies that predispose for EBV and KSHV infection do not identify crucial roles for cDCs or pDCs in the initiation of immune control over B cell infection by these two tumor viruses, suggesting that either antigen-presenting cell functions are redundant in these infections or that T cell priming is primarily performed by EBV- and KSHV-infected B cells.

## 4. Harnessing Dendritic Cells for Therapies against γ-Herpesvirus Associated Diseases 

This does, however, not mean that DCs cannot be harnessed therapeutically against the two oncogenic γ-herpesviruses. Along these lines, DCs are explored for both in vitro expansion of virus-specific T cell lines that then can be transferred into patients with EBV associated diseases and for targeting vaccine formulations to these potent antigen-presenting cells [35,36,37,38] (Figure 1). Indeed, adoptive T cell transfer has now shown, for more than 25 years, clinical benefits in patients that develop EBV-driven post-transplant lymphoproliferative disease (PTLD) after iatrogenic immune suppression to prevent transplant rejection [39]. At the time, EBV-specific T cells were expanded in vitro by restimulation with autologous LCLs prior to infusion. This protocol required time-consuming generation of the autologous LCLs, which were then used for the in vitro expansion of T cell lines without precisely defined antigen specificities. Therefore, expansion of T cell lines for adoptive transfer into patients with EBV associated malignancies that only express a subset of EBV antigens such as Hodgkin lymphoma and nasopharyngeal carcinoma was performed with antigen-loaded DCs [40,41]. For this purpose, primarily adenoviral infection of DCs to express the latent membrane proteins 1 and 2 (LMP1 and LMP2) or polyepitopes thereof with or without additional EBV nuclear antigen 1 (EBNA1) expression was used (Figure 1). Both stimulation protocols, as well as direct selection of EBNA1 specific T cells, demonstrated clinical efficacy [40,41,42]. Thus, adenovirus encoded EBV antigens (EBNA1, LMP1, and LMP2) presented by DCs can efficiently expand and even prime T cell lines, usually composed of both CD4^+^ and CD8^+^ T cells, that eliminate EBV associated tumors in patients. Similarly, DCs have been used to stimulate CD8^+^ T cells against the KSHV antigens gB, K8.1, LANA, and K12 from healthy KSHV carriers and might be used to generate T cell lines for adoptive transfer into patients with KSHV associated malignancies [43]. However, even so, monocyte-derived DCs and even dermal dendritic as well as Langerhans cells can be directly infected by KSHV [44,45], their immunogenicity, especially IL-12 production is attenuated, and thus directly KSHV-infected DCs might not be a suitable antigen-presenting cell population to optimally expand T cells for adoptive transfer into patients with Kaposi sarcoma or PEL [46].

In addition, EBV antigen targeting to DCs has been explored in preclinical models. For this purpose, EBNA1 and LMP1 have been coupled to the heavy chain of antibodies that target different receptors on DCs [20,36,38] (Figure 1). These reagents allow for robust EBV antigen processing for MHC class II-restricted presentation to CD4^+^ T cells, but only provide limited cross-presentation on MHC class I molecules to CD8^+^ T cells [38]. Such CD4^+^ T cell responses can also be elicited in vivo by targeting the human or mouse decalectin receptor DEC-205 in HIS mice or C57BL/6 mice [20,36,37,38]. The elicited CD4^+^ T cell responses correlate with antigen-specific antibody responses [36,38], and the respective CD4^+^ T cell clones can kill autologous LCLs in vitro [37]. DEC-205 expression is highest on CD141 positive cDC1s and allows these to take up DEC-205 targeted antigen most efficiently [37]. In contrast, CD8^+^ T cell responses against EBNA1 can not only in vitro, but also in vivo, be much more efficiently stimulated by adenoviral antigen expression [38]. Therefore, comprehensive CD4^+^ and CD8^+^ T cell priming and long-term maintenance of CD8^+^ T cell function require heterologous prime-boost vaccination with DEC-205 targeted and adenovirally expressed EBNA1 [38]. However, CD4^+^ T cell priming by DEC-205 targeted antigen can also be substituted by CD4^+^ T cell stimulation with modified vaccinia virus Ankara (MVA) [38]. Such comprehensive T cell responses are required to control EBNA1 expressing lymphomas in mice [38]. Thus, DCs can be harnessed to expand T cells for adoptive transfer therapy and EBV specific vaccination with recombinant viral vectors for CD8^+^ T cell induction and surface receptor targeting by antigen plus antibody hybrid molecules for CD4^+^ T cell response initiation. 

## 5. Conclusions

EBV and KSHV are two tumor viruses that are widely distributed in the human adult population and continuously threaten their carriers with oncogenesis. In response, the human immune system has learned during its long co-evolution with these viruses to control them to near perfection. Primarily cell-mediated immune control is required to keep EBV and KSHV in check. Curiously, very little evidence exists that DCs, the most potent antigen-presenting cells in the human body, are involved in the priming of this immune control. Indeed, both EBV and KSHV might have learned to steer clear of DC activation to establish persistence. Nevertheless, DCs can be harnessed to augment EBV-specific immune control either by harnessing them for expansion of virus-specific T cells that then can be adoptively transferred into patients with virus-induced tumors to eradicate their malignancies. Furthermore, approaches to target them for EBV- and KSHV-specific vaccination are explored. In light of these recent developments, however, antigen targeting to B cells should also be considered, which are probably at least involved in the expansion of γ-herpesvirus specific T cell responses to sufficient levels to control these oncogenic viruses.

## Figures and Tables

**Figure 1 viruses-11-01125-f001:**
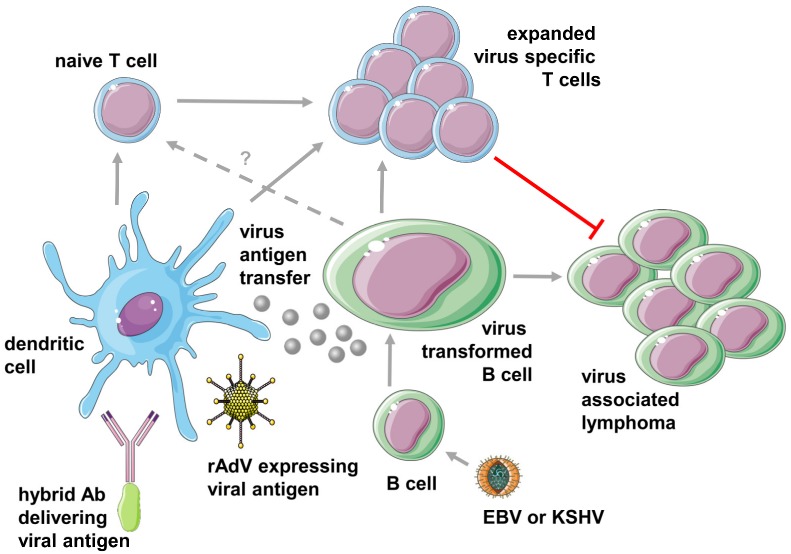
Dendritic cells in the initiation of EBV and KSHV specific T cell responses and treatments of pathologies that are associated with these viruses. EBV and KSHV primarily infect B cells within the hematopoietic lineage. EBV transforms these cells into potent antigen-presenting cells that are at least involved in the expansion of virus-specific T cells but might also prime these (dashed arrow). These expanded T cells then block the outgrowth of virus-transformed B cells into lymphomas. Dendritic cells receive virus antigens either via antigen transfer from infected cells or therapeutically via antibody plus antigen hybrid molecules as well as recombinant viruses, e.g., adenovirus (rAdV), for priming and expansion of virus-specific T cells. This figure was created in part with modified Servier Medical Art templates, which are licensed under a Creative Commons Attribution 3.0 Unported License: https://smart.servier.com.

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
