# Peer review of "The Role of Dendritic Cells in Immune Control and Vaccination against γ-Herpesviruses"

_viruses, 2019, doi:10.3390/v11121125_

Round 1

Reviewer 1 Report

This review on the role of DC in the development of immune responses to two gamma herpesviruses  is lacking some important references  to published work in the KSHV field.

In particular, in section 3 there is no mention of work by Lepone et al. (Clin. Vaccine Immunology 17: 1507, 2010 )on the role of cDC in stimulation KSHV-specific cytotoxic T cell responses to the virus lytic and latency antigens and the induction  of polyfunctional T cells.

In section 4, most of the writing is dedicated to the EBV interaction with DC, but no mention of two seminal papers describing the effect of KSHV virus infection of cDC and skin DC, namely Langerhans cells and dermal dendritic cells and the effect on T cell stimulation (Rappocciolo G. et al. J. Immunol 176: 1741, 2006; Rappocciolo G. et al. J.Virol. 91:, 2017) and the effect on cytokine production by KSHV infected DC ( Hensler et al. 90:79, 2009).

Minor corrections:

Line 72: need to define cross-presentation.

Section 3, Paragraph  line 105-108 is confusing

Author Response

This review on the role of DC in the development of immune responses to two gamma herpesviruses is lacking some important references to published work in the KSHV field.

I thank this reviewer for his/her suggestions regarding DC infection by and utilization against KSHV, which I have all included in the revised manuscript version.

In particular, in section 3 there is no mention of work by Lepone et al. (Clin. Vaccine Immunology 17: 1507, 2010) on the role of cDC in stimulation KSHV-specific cytotoxic T cell responses to the virus lytic and latency antigens and the induction  of polyfunctional T cells.

This reference has now been cited and discussed on page 8 of the revised manuscript.

In section 4, most of the writing is dedicated to the EBV interaction with DC, but no mention of two seminal papers describing the effect of KSHV virus infection of cDC and skin DC, namely Langerhans cells and dermal dendritic cells and the effect on T cell stimulation (Rappocciolo G. et al. J. Immunol 176: 1741, 2006; Rappocciolo G. et al. J.Virol. 91:, 2017) and the effect on cytokine production by KSHV infected DC ( Hensler et al. 90:79, 2009).

These three references have now been cited and discussed on page 8 of the revised manuscript.

Reviewer 2 Report

The author summmarized the role of denditic cells during infection with Epstein Barr virus and Kaposi sarcoma associated herpesvirus. Both viruses are important human pathogens and belong to the human gamma-herpesviruses. The review is concise and well written. From my point of view, there are only minor issues that have to be addressed.

Point 1: A reference to figure 1 should be given in the main text.

Point 2: in line 10 to line 11 "...most carriers’ immune control them for life." should probably read "...most carriers’ immune system control them for life." 

Point 3: in line 168-170 "...by harnessing them for virus specific T cell expansion that then can be adoptively transferred into patients..." should probaly read "...by harnessing them for expansion of virus specific T cells that then can be adoptively transferred into patients..." 

Author Response

The author summarized the role of dendritic cells during infection with Epstein Barr virus and Kaposi sarcoma associated herpesvirus. Both viruses are important human pathogens and belong to the human gamma-herpesviruses. The review is concise and well written. From my point of view, there are only minor issues that have to be addressed.

I have incorporated all of this reviewer’s minor suggestions into the revised manuscript.

Point 1: A reference to figure 1 should be given in the main text.

Figure 1 is now cited on page 8 of the revised manuscript.

Point 2: in line 10 to line 11 "...most carriers’ immune control them for life." should probably read "...most carriers’ immune system control them for life."

The respective sentence has been changed accordingly.

Point 3: in line 168-170 "...by harnessing them for virus specific T cell expansion that then can be adoptively transferred into patients..." should probaly read "...by harnessing them for expansion of virus specific T cells that then can be adoptively transferred into patients..."

The respective sentence has been changed accordingly.